# DETECTING ADVERSARIAL EXAMPLES

## ABSTRACT

Deep Neural Networks (DNNs) have been shown to be vulnerable to adversarial examples. While numerous successful adversarial attacks have been proposed, defenses against these attacks remain relatively understudied. Existing defense approaches either focus on negating the effects of perturbations caused by the attacks to restore the DNNs' original predictions or use a secondary model to detect adversarial examples. However, these methods often become ineffective due to the continuous advancements in attack techniques. We propose a novel universal and lightweight method to detect adversarial examples by analyzing the layer outputs of DNNs. Through theoretical justification and extensive experiments, we demonstrate that our detection method is highly effective, compatible with any DNN architecture, and applicable across different domains, such as image, video, and audio.

## 1 INTRODUCTION

Goodfellow et al. (2014) demonstrated that deep neural networks (DNNs) are vulnerable to adversarial examples and proposed the Fast Gradient Sign Method (FGSM) to craft these adversarial examples by adding perturbations to the model inputs, leveraging the linear nature of DNNs. After the initial introduction of FGSM, various adversarial attacks were proposed across different domains. However, compared to the vast diversity among attack techniques, existing defense methods are built on a few different strategies.

The most commonly used defense strategy is trying to remove the perturbations or altering the inputs to reduce their effects. Another and the oldest defense strategy is adversarial training where the training set of model contains adversarial examples.Both strategies do not generalize well to unseen attacks as they are tailored for a specific set of attacks. On the other hand, a recent defense strategy suggests to use a baseline method and compare its output with the main model to detect attacks. This strategy makes the defense vulnerable to the limitations of the baseline model and the potential mismatches between the main and baseline models for clean samples.

As opposed to the existing defense strategies which either analyze the model inputs or outputs, we propose Layer Regression (LR), a universal lightweight adversarial example detector, which analyzes the changes in the DNN's internal layer outputs. LR is highly effective for defending various DNN models against a wide range of attacks in different data domains such as image, video, and audio. LR utilizes the difference between the impacts of adversarial samples on early and final layers by performing regression among them. We present the following contributions:

- We propose the first universal defense strategy against adversarial examples, which take advantage of the sequential layer-based nature of DNNs and the common objectives of attacks.

- We conduct extensive experiments with 672 distinct attack-dataset-model-defense combinations for image recognition, and show that LR outperforms existing methods with a 97.6% average detection performance while the next best performance achieved by the existing defenses is 82.9%. Universality of LR is shown with its superior performance in detecting action recognition attacks and speech recognition attacks.

- In addition to its high performance across a wide range of domains, models, and attacks, LR is also the most lightweight defense method. It is orders of magnitude faster than the existing defenses, making it ideal for real-time attack detection.

## 2 RELATED WORKS

### 2.1 ADVERSARIAL ATTACKS

The robustness of deep neural networks and their vulnerability against adversarial examples have been investigated since the introduction of FGSM (Goodfellow et al., 2014). Numerous adversarial attacks have been proposed to generate effective adversarial examples in recent years (Madry et al., 2017; Croce & Hein, 2020; Kurakin et al., 2018; Chen et al., 2017; Ilyas et al., 2018; Mumcu & Yilmaz, 2024a; Wang & He, 2021; Fang et al., 2024; Gao et al., 2020). There are two main adversarial attack settings, namely white-box and black-box. While it is assumed that the attacker has access to the target model in the white-box setting, in the black-box setting, the attacker does not have any prior information about the target model.

White box attacks, including FGSM (Goodfellow et al., 2014), PGD (Madry et al., 2017), APGD (Croce & Hein, 2020), generate adversarial examples by maximizing the target model's loss function and they are usually referred as gradient-based attacks. BIM (Kurakin et al., 2018) tries to improve gradient-based attack by applying perturbations iteratively. Transferability based black-box attacks, which is introduced in Papernot et al. (2017), are one of the most common black-box approach. The idea is to use an attack on a substitute model for generating adversarial examples for unknown target models, utilizing the transferability of adversarial examples to different DNNs. While adversarial examples generated by these attacks are most effective when the substitute model exactly matches the target model as in a white-box attack setting, their success is shown to be transferable even when there is significant architectural differences between the substitute and target models. Wang & He (2021) introduced VMI and VNI to further extend iterative gradient-based attacks and try to achieve high transferability by considering the gradient variance of the previous iterations. PIF (Gao et al., 2020) uses patch-wise iterations to achieve transferability. ANDA (Fang et al., 2024) aims to achieve strong transferability by avoiding the overfitting of adversarial examples to the substitute model.

### 2.2 ADVERSARIAL DEFENSES

Attempting to make changes on the input data for removing the effects of perturbations from adversarial examples is the most common defense strategy. JPEG compression is studied in several works (Cucu et al., 2023; Aydemir et al., 2018; Das et al., 2018), and it is shown that compressing and decompressing helps to remove of adversarial effects on input images. Xie et al. (2017) uses random resizing and padding on the inputs to eliminate the adversarial effects. Xu (2017b) introduces feature squeezing where they use bit reduction, spatial smoothing, and non-local means denoising to detect adversarial examples. Several denoising methods (Liao et al., 2018; Xiong et al., 2022; Salman et al., 2020) were proposed to remove adversarial perturbations from the inputs. Mustafa et al. (2019) uses wavelet denoising and image super resolution as pre-processing steps to create a defense pipeline against adversarial attacks. Prakash et al. (2018) tries to eliminate adversarial effects by redistributing the pixel values via a process called pixel deflection. Adversarial training is another method which is studied to increase robustness of DNNs, however adversarial training often fails to perform well especially under various attack configurations (Bai et al., 2021). In addition, a recent defense strategy which utilizes a baseline model, e.g., Vision Language Models (VLMs), with the assumption that the output of target model and baseline model are close to each other for clean input, but are far away from each other for adversarital input. A recent example is demonstrated by Mumcu & Yilmaz (2024c) where they used CLIP (Radford et al., 2021) to detect adversarial video examples.

Adversarial training suffers from the increasing number of adversarial attacks which makes this strategy ineffective against adversarial examples which are not represented in the training set. Similarly, developing a universal perturbation removal method that is effective against every attack is not feasible due to continuous advancements in attack techniques. Additionally, since the defender cannot always know whether the input has been attacked, there is a risk of degrading clean inputs and causing false alarms. The baseline methods, which involve using a secondary model, such as VLMs, to detect adversarial examples, rely on the assumption that the secondary model will not be affected by the adversarial samples that are designed against the primary model. However, even without an attack, a mismatch between the models' predictions may trigger false alarms. Moreover, an attack that compromises both models can easily bypass this defense strategy.

## 3 DETECTING ADVERSARIAL EXAMPLES

Consider a Deep Neural Network (DNN) model $g(\cdot)$ that takes an input $x$ and predicts the target variable $y$ with $g(x)$. As discussed in Section 2.2, there are three main defense strategies against adversarial attacks: adversarial training, modifying input, and detecting adversarial samples by monitoring changes in output with respect to a baseline. While the former two focuses on the changes in the input ($x$ vs. $x^{adv}$), the latter utilizes the changes in the output ($g(x)$ vs. $g(x^{adv})$). Our approach differs from these existing approaches by leveraging the changes within the DNN activations. Instead of analyzing the input $x^{adv}$ or the output $g(x^{adv})$, the proposed defense method analyzes the intermediate steps between $x^{adv}$ and $g(x^{adv})$.

To develop a universal detector that work with any DNN and against any attack, we start with the following generic observation. Although there are numerous attacks with different approaches to generate adversarial examples, all attacks essentially aim to change the model's prediction by maximizing the loss $\mathcal{L}$ (e.g., cross-entropy loss) between prediction $g(x^{adv})$ and the ground truth $y$ while limiting the perturbation Fang et al. (2024):

$$\max_{x^{adv}} \mathcal{L}(g(x^{adv}), y) \text{ s.t. } \|x^{adv} - x\|_\infty \le \epsilon. \tag{1}$$

Considering this common aim of attack methods and the sequential nature of DNNs, in the following theorem, we show that the impact of adversarial examples on the final layer is much higher than the initial layer. Let us first define a generic DNN $g(\cdot)$ consisting of $n$ layers $a = \{a_1, a_2, ..., a_n\}$. In DNNs, including CNNs, transformers, etc., layers incrementally process the information from the previous layers to compute their respective outputs to the next layer. For example, for a model $g$ where each layer is connected to the previous one, the final output of the model can be formulated as follows:

$$g(x) = a_n(a_{n-1}(\ldots a_1(x))). \tag{2}$$

For simplicity, we will denote a layer's output vector with $a_i(x)$. Note that, the output of last layer $a_n(x) = g(x)$ is typically the class probability vector in classification, and $a_{n-1}(x)$ is referred as the feature vector of the model.

**Theorem 1.** *Assuming a loss function $\mathcal{L}(g(x), y)$ that is monotonic with $\|g(x) - y\|_\infty$, for $n > 1$, we have*

$$d_n = \|a_n(x^{adv}) - a_n(x)\|_\infty > d_1 = \|a_1(x^{adv}) - a_1(x)\|_\infty. \tag{3}$$

*Proof.* Since the model $g$ is trained by optimizing the weights $w$ to minimize the loss $\mathcal{L}(g_w(x), y)$, i.e.,

$$g(x) = \arg\min_w \mathcal{L}(g_w(x), y), \tag{4}$$

we can rewrite Eq. 1 as

$$\max_{x^{adv}} \mathcal{L}(g(x^{adv}), g(x)) \text{ s.t. } \|x^{adv} - x\|_\infty \le \epsilon. \tag{5}$$

Note that $a_n(x) = g(x)$ and $a_n(x^{adv}) = g(x^{adv})$ and $\mathcal{L}(g(x^{adv}), g(x))$ is monotonic with $\|g(x^{adv}) - g(x)\|_\infty$. Hence, $\|a_n(x^{adv}) - a_n(x)\|_\infty$ is maximized while limiting $\|x^{adv} - x\|_\infty$ by a small number. Finally, the perturbation aligned with DNN weights is amplified as it sequentially moves through the DNN layers Goodfellow et al. (2014),

$$\|x^{adv} - x\|_\infty < \|a_1(x^{adv}) - a_1(x)\|_\infty < \|a_n(x^{adv}) - a_n(x)\|_\infty, \ n > 1. $$

$\square$

Having shown that the impact of an adversarial sample is higher on the final layer output than the first layer output, we next present how to utilize this fact for detecting adversarial samples.

**Corollary 1.** *Consider a function $f$ which maps $a_1(x)$ to $a_n(x)$ and is stable in the sense that $\|f(a_1(x)) - f(a_1(x + \epsilon))\|_\infty \le \delta$ for small $\epsilon$ and $\delta$. Then,*

$$e_a = \|f(a_1(x^{adv})) - a_n(x^{adv})\|_\infty > e_c = \|f(a_1(x)) - a_n(x)\|_\infty. \tag{6}$$

*Proof.* Since $\|x^{adv} - x\|_\infty \le \epsilon$ from Eq. 1, $f(a_1(x^{adv}))$ is close to $f(a_1(x))$ due to the stability of $f$. From Theorem 1, $a_n(x^{adv})$ is far away from $a_n(x)$ compared to the distance between $a_1(x)$ and $a_1(x^{adv})$. Thus, the estimation error for adversarial samples $e_a = \|f(a_1(x^{adv})) - a_n(x^{adv})\|_\infty$ is larger than the error for clean samples $e_c = \|f(a_1(x)) - a_n(x)\|_\infty$. $\square$

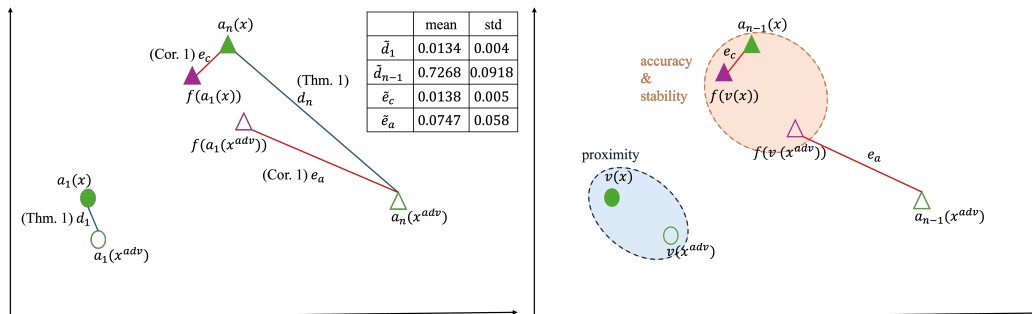

Figure 1: (Left) Theorem 1 shows that the impact of adversarial samples is higher on the final layer than the first layer. Corollary 1 uses this result to prove that the error of a stable estimator is higher for adversarial samples compared to clean samples. (Right) The performance of approximations in proposed detector to Corollary 1 depend on two conflicting objectives: proximity of input vectors for clean and adversarial samples, and training an accurate and stable estimator. (Table) Empirical confirmation of the agreement between the proposed detector with approximations and the theoretical results.

## 3.1 Layer Regression

While Theorem 1 provides the theoretical motivation, the result in Corollary 1 provides a mechanism to detect adversarial samples if we can train a suitable function $f$ (Figure 1). We make four approximations to obtain a practical algorithm based on Corollary 1. First, we propose to use a multi-layer perceptron (MLP) to approximate $f$. Second, since $a_n(x)$ denotes the predicted class probabilities, we choose the feature vector $a_{n-1}$, which takes unconstrained real values, as the target to train MLP as a regression model. Third, we approximate non-differentiable $\|\cdot\|_\infty$ with $\|\cdot\|_2$ to train the MLP using the differentiable mean squared error (MSE) loss. To empirically check the validity of Theorem 1 under these three approximations, we use the ImageNet validation dataset, ResNet-50 He et al. (2015) as target model, and PGD attack Madry et al. (2017) to compute the mean and standard deviation of normalized change in layer 1 $\tilde{d}_1 = \frac{\|a_1(x^{adv})-a_1(x)\|_2}{\|a_1(x^{adv})\|_2+\|a_1(x)\|_2}$ and layer $n-1$ $\tilde{d}_{n-1} = \frac{\|a_{n-1}(x^{adv})-a_{n-1}(x)\|_2}{\|a_{n-1}(x^{adv})\|_2+\|a_{n-1}(x)\|_2}$.

In deep neural networks with $n \gg 1$, training an suitable $f$ to estimate the feature vector $a_{n-1}$ using the first layer output $a_1$ as the input is a challenging task due to the highly nonlinear mapping in $n-2$ layers. To develop a lightweight detector via MLP, as the fourth approximation, we propose selecting a mixture of early layer outputs as the input to the regression model instead of using only the first layer. Using a mixture of 5th, 8th, and 13th convolutional layers in ResNet-50 as the input, we empirically check Corollary 1 under the same setting used for $\tilde{d}_1$ and $\tilde{d}_{n-1}$ by computing the mean and standard deviation of MSE $\tilde{e}_c = \|f(a_1(x)) - a_{n-1}(x)\|_2$ for clean images and $\tilde{e}_a = \|f(a_1(x^{adv})) - a_{n-1}(x^{adv})\|_2$ for adversarial images, where an MLP with two hidden layers is used for $f$. Results shown in Figure 1 corroborate Theorem 1 and Corollary 1 under the three approximations. Input selection for MLP is further discussed in this section and ablation study in Section 5.2.

Utilizing the four approximations to Corollary 1 discussed above, we propose a universal and lightweight detection algorithm with the following steps: (i) select a subset of the first $n-2$ layer vectors and generate a new vector $v$ from the selected subset, (ii) feed $v$ into a regression model $m$ to predict the feature vector $a_{n-1}(x)$, (iii) train $m$ by minimizing the mean squared error (MSE) loss $\ell(m(v), a_{n-1}(x))$ in the clean training set devoid of adversarial samples, (iv) decide a test sample is adversarial if the prediction loss is greater than a threshold, $\ell(m(v), a_{n-1}(x)) > h$. A pseudo-code of LR is given in Appendix D.

The formation of vector $v$ can be done in various ways, such as using only the $i$th layer vector $v = a_i(x)$ or mixture of several layer vectors. To enable larger estimation error $e_a$ for adversarial samples than estimation error $e_c$ for clean samples, the choice for $v$ should strike a balance between two competing goals, as illustrated in Figure 1: proximity of clean $v(x)$ and adversarial $v(x^{adv})$, and accuracy and stability of estimation function $f$. While training an accurate and stable regression

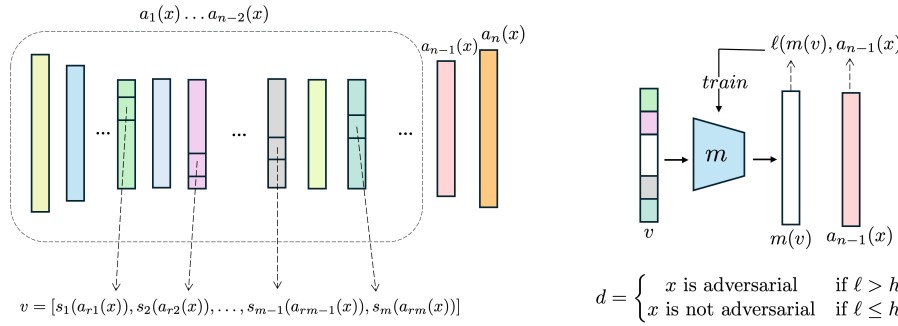

Figure 2: (Left) Layer selection and slicing operations to form the input vector. (Right) Training and testing procedures of the proposed detector.

model is more feasible when $v$ is selected from the layers closer to the target layer $n-1$, e.g., $v = a_{n-2}(x)$, such a detector might be less sensitive to adversarial samples since both $a_{n-2}(x)$ and $a_{n-1}(x)$ are expected to be impacted significantly by the attack, i.e., $a_{n-2}(x)$ and $a_{n-2}(x^{adv})$ will not be proximal. On the other hand, while selecting $v = a_1(x)$ ensures a reasonably small perturbation in $v$, it also makes obtaining an accurate and stable estimator more challenging. As a result, we propose to select a subset of layer vectors

$$a_r = \{a_{r1}(x), a_{r2}(x), ..., a_{rm}(x)\} \tag{7}$$

where $a_r \in a$ and $m < n$ is the number of selected layers. From the selected layer vectors, we aim to generate a new vector $v$. However, since the layer vectors are often large due to the operations like convolutions or attentions, to get a specific portion of the selected layer vectors, we define a unique slicing function $s = \{s_1, s_2, ..., s_m\}$ for each layer vector in $a_r$. Then, each slicing function is applied to the corresponding layer vector in $a_r$ to get the sliced vectors

$$s_r = \{s_1(a_{r1}(x)), s_2(a_{r2}(x)), ..., s_m(a_{rm}(x))\}. \tag{8}$$

Finally, the vector $v$ is generated by concatenating the vectors in $s_r$:

$$v = [s_1(a_{r1}(x)), s_2(a_{r2}(x)), \ldots, s_m(a_{rm}(x))]. \tag{9}$$

The proposed layer selection and slicing process is summarized in Figure 2.

During the training, only the clean input samples are used. After the training, the loss is expected to be low for clean inputs and high for adversarial inputs. Thus, an adversarial example can be detected by comparing the loss $\ell(m(v), a_{n-1}(x))$ with a threshold $h$:

$$d = \begin{cases} x \text{ is adversarial} & \text{if } \ell > h \\ x \text{ is not adversarial} & \text{if } \ell \leq h, \end{cases} \tag{10}$$

The threshold $h$ is determined by calculating the loss for a set of clean inputs $\beta = \{\ell_1, \ldots, \ell_K\}$, where the losses in $\beta$ are sorted in ascending order, i.e., $\ell_1 \leq \ell_2 \leq \cdots \leq \ell_K$. $h$ is selected as the $\theta$th percentile of the clean training losses: $h = \beta[\lfloor K\theta/100 \rfloor]$ where $\lfloor \cdot \rfloor$ denotes the floor operator, and $\beta[i]$ denotes the $i$th element of $\beta$.

## 4 EXPERIMENTS AND ANALYSES

In this section, we evaluate the performance of our method and compare it against 7 existing defense methods. Our extensive experiments are conducted on 2 widely used image datasets, with 6 image classification models and 7 adversarial attacks.

**Defenses:** The following defense methods are used in the experiments: JPEG compression (JPEG) (Das et al., 2018), Randomization (Random)(Xie et al., 2017), Deflection (Deflect) (Prakash et al., 2018), VLAD (Mumcu & Yilmaz, 2024c), Feature Squeezing (FS) Xu (2017a), Wavelet denoising (Denoise) and super resolution (WDSR) (Mustafa et al., 2019). Since VLAD and FS were proposed as detectors in their respective papers, we use their official implementations. The remaining methods were proposed to increase robustness by altering the inputs to remove perturbations. For this reason,

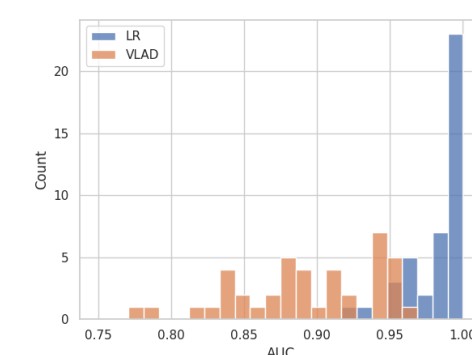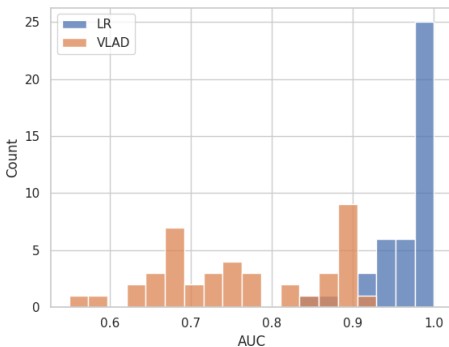

Figure 3: Histogram of AUC values for LR and VLAD on (Left) ImageNet (Right) CIFAR-100.

we obtain a detection method from them by comparing the predictions after and before the methods are applied. The case the predictions match indicates no attack and a mismatch indicates an attack.

**Attacks:** BIM (Kurakin et al., 2018), PGD (Madry et al., 2017), PIF (Gao et al., 2020), APGD (Croce & Hein, 2020), ANDA (Fang et al., 2024), VMI and VNI (Wang & He, 2021) attacks are used to generate adversarial examples during the experiments. We consider the most challenging case for the detector, the white-box attack scenario, in which the attacker has full access to the classification model, i.e., no mismatch between the surrogate model and actual model.

**Models:** To represent the different architectures we used three CNN-based image classification models, VGG19 Simonyan & Zisserman (2014), ResNet50 He et al. (2015), Inceptionv3 Szegedy et al. (2015); and three transformers-based models, ViT Dosovitskiy (2020), DeiT Touvron et al. (2021), LeViT Graham et al. (2021).

**Datasets**: The experiments are conducted using the validation set of ImageNet dataset (Russakovsky et al., 2015) and the CIFAR-100 dataset (Krizhevsky et al., 2009). For testing, the clean set comprises all images that are correctly classified by the target models. An adversarial set is formed for each attack-target model combination by gathering the attack's adversarial images misclassified by target model.

**LR training:** For each model, an MLP with 2 hidden layers is trained as the LR detector. The details of the chosen $a_r$ subsets, detectors, $s_r$ slicing functions, and training parameters are detailed in Appendix C.

**Evaluation:** The commonly used the Area Under Curve (AUC) metric from the Receiver Operating Characteristic (ROC) curve is used to evaluate the attack detection performance of defense methods. ROC curve shows the trade-off between true positive rate (i.e., ratio of successfully detected adversarial samples to all adversarial samples) and false positive rate (i.e., ratio of false alarms to all clean samples).

## 4.1 AUC RESULTS

In Table 1, we report the AUC scores for JPEG, Random, Deflect, VLAD, Denoise, WDSR, FS and our method, LR, against the BIM, PGD, PIF, APGD, ANDA, VMI and VNI attacks targeting 7 models, namely Vgg19, ResNet50, Inceptionv3, ViT, DeiT, LeViT, on two datasets, ImageNet and CIFAR-100. In every experimental setting, with different attacks, target models, and datasets, our proposed method outperforms the existing defense methods by a wide margin. Compared to LR's average AUC of 0.976 over ImageNet and CIFAR, the best performance among the existing methods remains at 0.829. More importantly, our method is robust to changing targets and attacks, as indicated by its small standard deviation. While the other defense methods are effective against certain target-attack combinations, they fail to generalize this to a wide range of settings. For instance, on ImageNet, JPEG, Random, and FS perform better with the transformer models, however they rarely exceed the random guess performance with the CNN models. While VLAD consistently achieves good performance on ImageNet, its performance significantly drops on CIFAR-100, possibly due to

| | | ImageNet | | | | | | | | CIFAR-100 | | | | | | | |
|---|---|---|---|---|---|---|---|---|---|---|---|---|---|---|---|---|---|
| | | JPEG | Random | Deflect | VLAD | Denoise | WDSR | FS | Ours | JPEG | Random | Deflect | VLAD | Denoise | WDSR | FS | Ours |
| Vgg19 | BIM | 0.424 | 0.427 | 0.482 | 0.950 | 0.451 | 0.393 | 0.130 | **0.999** | 0.468 | 0.314 | 0.490 | 0.687 | 0.481 | 0.411 | 0.006 | **0.999** |
| | PGD | 0.496 | 0.457 | 0.482 | 0.950 | 0.454 | 0.463 | 0.222 | **0.998** | 0.483 | 0.313 | 0.490 | 0.752 | 0.482 | 0.439 | 0.018 | **0.999** |
| | PIF | 0.328 | 0.443 | 0.482 | 0.952 | 0.450 | 0.314 | 0.078 | **0.999** | 0.466 | 0.314 | 0.490 | 0.867 | 0.481 | 0.408 | 0.005 | **0.999** |
| | APGD | 0.530 | 0.463 | 0.482 | 0.964 | 0.454 | 0.487 | 0.229 | **0.998** | 0.482 | 0.316 | 0.490 | 0.837 | 0.481 | 0.423 | 0.023 | **0.998** |
| | ANDA | 0.392 | 0.474 | 0.491 | 0.946 | 0.467 | 0.380 | 0.264 | **0.949** | 0.482 | 0.359 | 0.492 | 0.666 | 0.487 | 0.425 | 0.113 | **0.994** |
| | VMI | 0.356 | 0.418 | 0.482 | 0.940 | 0.451 | 0.342 | 0.111 | **0.999** | 0.467 | 0.322 | 0.490 | 0.667 | 0.482 | 0.409 | 0.007 | **0.999** |
| | VNI | 0.405 | 0.448 | 0.484 | 0.956 | 0.461 | 0.382 | 0.190 | **0.997** | 0.474 | 0.318 | 0.491 | 0.740 | 0.483 | 0.415 | 0.022 | **0.999** |
| ResNet50 | BIM | 0.670 | 0.633 | 0.493 | 0.839 | 0.482 | 0.546 | 0.293 | **0.989** | 0.820 | 0.687 | 0.455 | 0.778 | 0.618 | 0.728 | 0.387 | **0.981** |
| | PGD | 0.771 | 0.745 | 0.493 | 0.846 | 0.485 | 0.676 | 0.533 | **0.984** | 0.874 | 0.707 | 0.452 | 0.774 | 0.714 | 0.821 | 0.399 | **0.994** |
| | PIF | 0.510 | 0.644 | 0.493 | 0.841 | 0.482 | 0.465 | 0.288 | **0.963** | 0.725 | 0.715 | 0.449 | 0.756 | 0.477 | 0.609 | 0.626 | **0.967** |
| | APGD | 0.746 | 0.703 | 0.493 | 0.856 | 0.485 | 0.640 | 0.467 | **0.968** | 0.838 | 0.646 | 0.457 | 0.782 | 0.616 | 0.778 | 0.435 | **0.961** |
| | ANDA | 0.450 | 0.484 | 0.494 | 0.787 | 0.484 | 0.444 | 0.135 | **0.959** | 0.556 | 0.530 | 0.448 | 0.675 | 0.478 | 0.448 | 0.068 | **0.994** |
| | VMI | 0.493 | 0.519 | 0.493 | 0.815 | 0.484 | 0.445 | 0.140 | **0.990** | 0.613 | 0.581 | 0.451 | 0.718 | 0.497 | 0.499 | 0.231 | **0.992** |
| | VNI | 0.531 | 0.536 | 0.494 | 0.826 | 0.491 | 0.470 | 0.212 | **0.967** | 0.616 | 0.549 | 0.453 | 0.737 | 0.499 | 0.513 | 0.277 | **0.965** |
| InceptionV3 | BIM | 0.549 | 0.534 | 0.496 | 0.882 | 0.492 | 0.768 | 0.235 | **0.981** | 0.877 | 0.802 | 0.428 | 0.904 | 0.735 | 0.806 | 0.363 | **1.000** |
| | PGD | 0.615 | 0.576 | 0.497 | 0.876 | 0.494 | 0.796 | 0.356 | **0.965** | 0.861 | 0.662 | 0.425 | 0.898 | 0.764 | 0.807 | 0.324 | **0.999** |
| | PIF | 0.484 | 0.518 | 0.496 | 0.881 | 0.488 | 0.667 | 0.107 | **0.993** | 0.821 | 0.889 | 0.443 | 0.901 | 0.574 | 0.721 | 0.344 | **1.000** |
| | APGD | 0.604 | 0.564 | 0.497 | 0.894 | 0.496 | 0.794 | 0.361 | **0.955** | 0.880 | 0.817 | 0.434 | 0.895 | 0.771 | 0.806 | 0.376 | **0.999** |
| | ANDA | 0.490 | 0.499 | 0.498 | 0.841 | 0.493 | 0.515 | 0.228 | **0.920** | 0.713 | 0.571 | 0.427 | 0.811 | 0.504 | 0.591 | 0.292 | **0.999** |
| | VMI | 0.483 | 0.502 | 0.496 | 0.867 | 0.489 | 0.671 | 0.117 | **0.984** | 0.848 | 0.783 | 0.424 | 0.877 | 0.496 | 0.799 | 0.346 | **1.000** |
| | VNI | 0.519 | 0.527 | 0.497 | 0.886 | 0.498 | 0.703 | 0.249 | **0.958** | 0.860 | 0.787 | 0.434 | 0.884 | 0.600 | 0.806 | 0.311 | **1.000** |
| ViT | BIM | 0.872 | 0.896 | 0.533 | 0.940 | 0.640 | 0.817 | 0.928 | **0.996** | 0.500 | 0.657 | 0.499 | 0.721 | 0.499 | 0.492 | 0.287 | **0.980** |
| | PGD | 0.850 | 0.867 | 0.527 | 0.911 | 0.602 | 0.790 | 0.911 | **0.988** | 0.494 | 0.670 | 0.499 | 0.692 | 0.496 | 0.484 | 0.272 | **0.967** |
| | PIF | 0.790 | 0.824 | 0.517 | 0.891 | 0.512 | 0.669 | 0.865 | **0.996** | 0.489 | 0.532 | 0.499 | 0.697 | 0.494 | 0.476 | 0.150 | **0.982** |
| | APGD | 0.856 | 0.897 | 0.529 | 0.925 | 0.625 | 0.800 | 0.923 | **0.990** | 0.504 | 0.707 | 0.499 | 0.749 | 0.500 | 0.498 | 0.317 | **0.949** |
| | ANDA | 0.717 | 0.678 | 0.519 | 0.846 | 0.557 | 0.638 | 0.823 | **0.972** | 0.493 | 0.517 | 0.499 | 0.587 | 0.495 | 0.481 | 0.262 | **0.846** |
| | VMI | 0.766 | 0.811 | 0.513 | 0.908 | 0.572 | 0.701 | 0.857 | **0.995** | 0.489 | 0.575 | 0.499 | 0.691 | 0.494 | 0.477 | 0.153 | **0.991** |
| | VNI | 0.776 | 0.826 | 0.524 | 0.906 | 0.605 | 0.727 | 0.909 | **0.991** | 0.499 | 0.595 | 0.499 | 0.701 | 0.499 | 0.489 | 0.268 | **0.947** |
| DeiT | BIM | 0.859 | 0.889 | 0.524 | 0.955 | 0.575 | 0.784 | 0.884 | **0.998** | 0.496 | 0.637 | 0.499 | 0.669 | 0.494 | 0.481 | 0.359 | **0.915** |
| | PGD | 0.863 | 0.877 | 0.528 | 0.947 | 0.583 | 0.801 | 0.900 | **0.997** | 0.499 | 0.655 | 0.499 | 0.674 | 0.494 | 0.484 | 0.344 | **0.867** |
| | PIF | 0.774 | 0.835 | 0.512 | 0.926 | 0.504 | 0.711 | 0.620 | **0.999** | 0.484 | 0.520 | 0.498 | 0.654 | 0.491 | 0.473 | 0.158 | **0.972** |
| | APGD | 0.853 | 0.896 | 0.522 | 0.947 | 0.570 | 0.776 | 0.882 | **0.997** | 0.499 | 0.653 | 0.499 | 0.675 | 0.496 | 0.484 | 0.359 | **0.908** |
| | ANDA | 0.766 | 0.755 | 0.530 | 0.914 | 0.577 | 0.701 | 0.752 | **0.985** | 0.489 | 0.512 | 0.498 | 0.568 | 0.492 | 0.477 | 0.257 | **0.914** |
| | VMI | 0.785 | 0.837 | 0.509 | 0.946 | 0.528 | 0.685 | 0.797 | **0.999** | 0.488 | 0.562 | 0.499 | 0.627 | 0.493 | 0.477 | 0.299 | **0.957** |
| | VNI | 0.804 | 0.853 | 0.518 | 0.944 | 0.549 | 0.719 | 0.849 | **0.997** | 0.494 | 0.574 | 0.498 | 0.631 | 0.495 | 0.483 | 0.281 | **0.944** |
| LeViT | BIM | 0.679 | 0.731 | 0.501 | 0.888 | 0.502 | 0.621 | 0.640 | **0.993** | 0.929 | 0.834 | 0.684 | 0.891 | 0.884 | 0.855 | 0.995 | **0.999** |
| | PGD | 0.694 | 0.736 | 0.497 | 0.868 | 0.493 | 0.622 | 0.661 | **0.990** | 0.926 | 0.839 | 0.569 | 0.913 | 0.841 | 0.850 | 0.862 | **0.934** |
| | PIF | 0.537 | 0.647 | 0.497 | 0.878 | 0.486 | 0.489 | 0.508 | **0.991** | 0.855 | 0.817 | 0.495 | 0.831 | 0.533 | 0.800 | 0.620 | **0.934** |
| | APGD | 0.748 | 0.797 | 0.500 | 0.914 | 0.508 | 0.687 | 0.759 | **0.976** | 0.929 | 0.832 | 0.680 | 0.891 | 0.878 | 0.855 | 0.986 | **0.993** |
| | ANDA | 0.494 | 0.506 | 0.499 | 0.776 | 0.492 | 0.485 | 0.429 | **0.936** | 0.838 | 0.705 | 0.524 | 0.876 | 0.574 | 0.814 | 0.646 | **0.931** |
| | VMI | 0.527 | 0.586 | 0.496 | 0.840 | 0.487 | 0.482 | 0.387 | **0.997** | 0.925 | 0.833 | 0.658 | 0.891 | 0.661 | 0.856 | 0.995 | **0.999** |
| | VNI | 0.595 | 0.661 | 0.500 | 0.881 | 0.505 | 0.540 | 0.588 | **0.986** | 0.921 | 0.827 | 0.666 | 0.895 | 0.714 | 0.856 | 0.992 | **0.998** |
| Average | | 0.629 | 0.655 | 0.502 | 0.893 | 0.511 | 0.609 | 0.495 | **0.982** | 0.653 | 0.619 | 0.496 | 0.765 | 0.565 | 0.602 | 0.353 | **0.970** |
| Std | | 0.160 | 0.159 | 0.014 | 0.048 | 0.049 | 0.146 | 0.296 | 0.018 | 0.183 | 0.168 | 0.064 | 0.100 | 0.119 | 0.168 | 0.274 | 0.037 |

Table 1: Comparison between 8 defenses in terms of detection AUC against 7 attack methods targeting 6 models using ImageNet and CIFAR-100 datasets.

its dependency on CLIP. When CLIP is fooled by the attacks, VLAD also fails. Moreover, VLAD is orders of magnitude slower than LR, as shown in Figure 4 and Table 2.

The histogram of AUC values for LR in Figure 3 clearly demonstrate the robustness of our method compared to its main competitor, VLAD. ANDA, which is the most recent and state-of-the-art attack method in the literature, troubles our detector the most. However, LR still maintains an average AUC of $0.954$ and $0.946$ against ANDA over the six target models on ImageNet and CIFAR-100, respectively. The average AUC values of LR against BIM, PGD, PIF, APGD, VMI, and VNI on (ImageNet, CIFAR-100) are as follows: $(0.993, 0.979), (0.987, 0.960), (0.990, 0.976), (0.981, 0.968), (0.994, 0.990), (0.983, .976)$.
Interestingly, InceptionV3 is the easiest model for LR to defend against attacks trained on CIFAR-100 with a perfect detection score while the average AUC of LR for InceptionV3 with ImageNet is $0.965$. The average AUC of LR values for Vgg19, ResNet50, ViT, DeiT, and LeViT with (ImageNet, CIFAR-100) are as follows: $(0.991, 0.998), (0.974, 0.979), (0.990, 0.952), (0.996, 0.925), (0.981, 0.970)$.

## 4.2 REAL-TIME DETECTION PERFORMANCE

Real-time performance is a crucial aspect of an attack detection method. A detection mechanism must always run alongside the DNN model to ensure timely detection of adversarial examples. Here, we compare the real-time performance of the defense methods used in the experiments. For each defense method, we processed 1,000 samples from ImageNet with the defense methods and took the average of processing times. In Table 2, for each defense method, we show the processing time

| Defenses | PTS (sec) |
|----------|-----------|
| JPEG | 0.0029 |
| Random | 0.0026 |
| Deflect | 0.0035 |
| VLAD | 0.1431 |
| Denoise | 0.0057 |
| WDSR | 0.2611 |
| FS | 0.0382 |
| Ours | 0.0004 |

Figure 4: The proposed detector is both universal (high AUC across all scenarios) and lightweight (fastest among all existing defenses).

Table 2: Processing time per sample in seconds for defense methods.

per sample (PTS) in seconds. Our method is significantly faster than the others with only 0.0004 seconds. WDSR and VLAD are the slowest methods with 0.2611 and 0.1431 seconds respectively. Figure 4 demonstrates the trade-off between PTS and average accuracy of the existing defense methods. Our method exhibits an outlying performance by significantly beating the existing methods in both PTS and average detection AUC. The processing times are measured using a computer with an NVIDIA 4090 GPU, AMD Ryzen 9 7950X CPU, and 64 GB RAM.

## 5 ABLATION STUDY

### 5.1 EFFECT OF ATTACK STRENGTH

Adversarial attacks often use a designated parameter $\epsilon$ to adjust the amount of perturbation on adversarial examples. For the experiments in Section 4, we used the default attack settings that are proposed by the authors. In this ablation study, we investigate the performance of defense methods against PGD and BIM attacks with different $\epsilon$ values. In the original implementation of PGD and BIM, attack strength parameter $\epsilon$ is set to 0.03. In addition to the original $\epsilon$, we also generated attacks by setting $\epsilon$ to 0.01, 0.09, 0.12, 0.15 and 0.3 and attacked ResNet and ViT.

Figure 5 plots the performances of defense methods for ResNet-PGD, ResNet-BIM, ViT-PGD and ViT-BIM model-attack pairs. We noticed a small performance drop in our method when the attack strength has the smallest value of 0.01. However, higher $\epsilon$ values resulted in higher AUC in every case. This is an expected behaviour since our method depends on the effects of the perturbation on DNN layers. In other defense methods, while VLAD and Deflect experience only small changes under stronger attacks, remaining methods are greatly affected by the $\epsilon$ value, especially while defending ResNet.

### 5.2 EFFECTS OF LAYER VECTOR CHOICE

For the experiments in Section 4, we used three layer vectors to form $v$, i.e., $m = 3$ in Eq. 9. Since the number of combinations for layer selection and slicing yields an intractable search space for extensive optimization, we performed optimization over a limited set of representative options. Table 3 summarizes the results of a preliminary experiment on CIFAR-100, which is conducted with DeiT as the target model and PGD and PIF as the adversarial attacks. In this experiment, we considered 25 attention layers of the model and trained our detectors by forming $v$ in four different ways: (1) using only the first attention layer vector $a_1$, (2) using only the last attention layer vector $a_{25}$, (3) combining the vectors from fifth, sixth, and seventh attention vectors $[a_5, a_6, a_7]$, (4) and combining the vectors from eighth, thirteenth, and seventeenth attention vectors $[a_8, a_{13}, a_{17}]$. The best strategy turns out to combining vectors from early layers $[a_5, a_6, a_7]$ as it strikes a good balance in the trade-off between the two conflicting goals (Figure 1): proximity of $v(x)$ and $v(x^{adv})$, and accuracy and stability of estimator $f$. Additionally, having multiple layers and slicing functions can create randomness in each LR implementation, which makes developing an attack against our

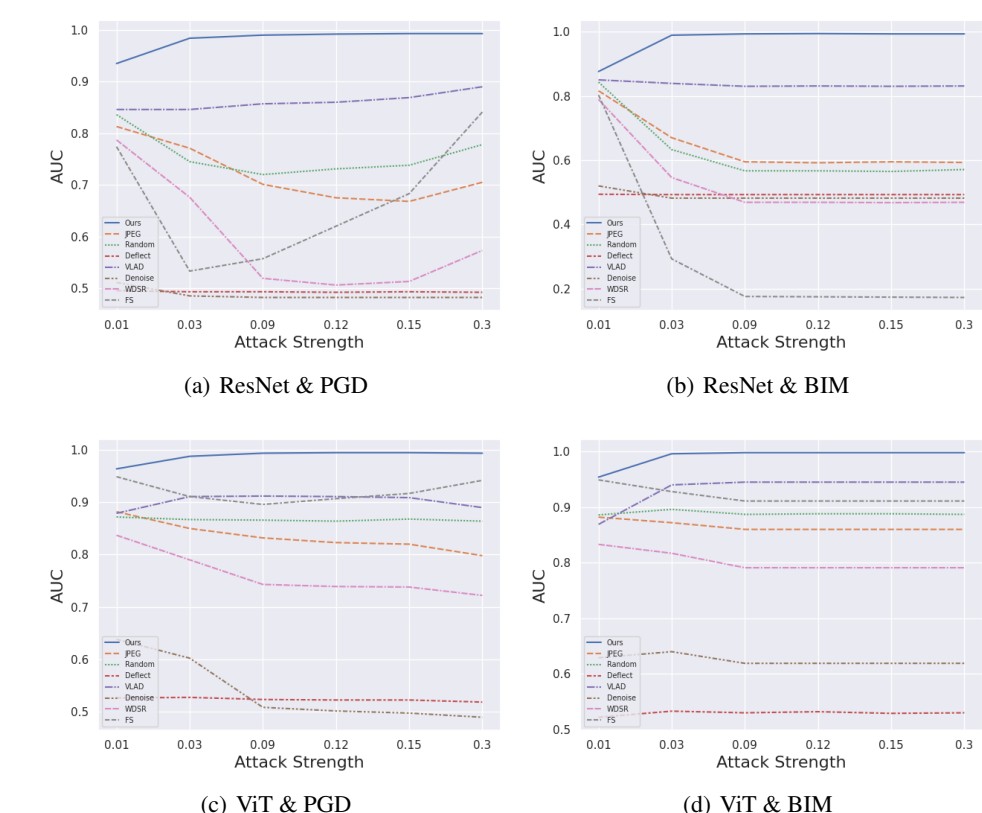

(a) ResNet & PGD  (b) ResNet & BIM

(c) ViT & PGD  (d) ViT & BIM

Figure 5: Impact of attack strength on detection performance.

| Attack | $a_1$ | $[a_5, a_6, a_7]$ | $[a_8, a_{13}, a_{17}]$ | $a_{n-2}$ |
|--------|-------|-------------------|-------------------------|-----------|
| PGD | 0.697 | 0.867 | 0.751 | 0.567 |
| PIF | 0.744 | 0.972 | 0.746 | 0.521 |

Table 3: Impact of layer selection on detection AUC.

detection system harder. The layer selection and slicing strategies that gave the best results reported in Table 1 are listed in Appendix C.

## 6 APPLICABILITY IN OTHER DOMAINS

LR is applicable in every domain where DNNs are used. In this section we demonstrate its performance in two other domains, video action recognition and speech recognition. In Appendices A and G, we provide more information about experimental details that are used in this section. Moreover, we provide additional results on another application, traffic sign recognition, in Appendix F.

### 6.1 LR FOR VIDEO ACTION RECOGNITION

In this section, we implement LR against video action recognition attacks and compare its performance with existing defenses designed for action recognition models, namely Advit (Xiao et al., 2019), Shuffle (Hwang et al., 2023), and VLAD (Mumcu & Yilmaz, 2024c). In the experiments, PGD-v attack (Mumcu & Yilmaz, 2024c) and Flick attack (Pony et al., 2021) are used to target MVIT (Fan et al., 2021) and X3D (Feichtenhofer, 2020). The experimental settings in Mumcu & Yilmaz (2024c) on Kinetics-400 (Kay et al., 2017) dataset are followed. The results in Table 4 show that LR outperforms the other defenses with an average AUC of 0.93%, followed by VLAD which achieves 0.91%.

|      |       | Advit | Shuffle | VLAD | LR (ours) |
|------|-------|-------|---------|------|-----------|
| MVIT | PGD-v | 0.93  | 0.98    | 0.93 | 0.99      |
|      | Flick | 0.34  | 0.65    | 0.87 | 0.89      |
| X3D  | PGD-v | 0.92  | 0.76    | 0.97 | 0.95      |
|      | Flick | 0.54  | 0.59    | 0.90 | 0.92      |
|      | Average | 0.68 | 0.74   | 0.91 | 0.93      |

Table 4: Comparison between defense methods in terms of detection AUC against attack methods targeting video action recognition models.

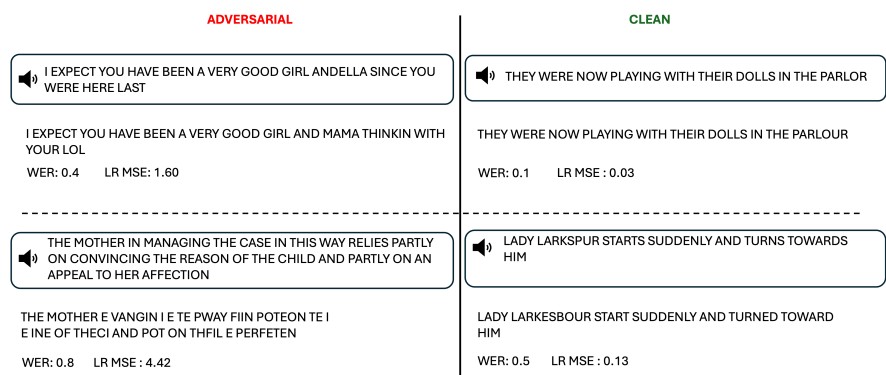

Figure 6: Clean and adversarial samples for speech recognition with ground truth in box, recognized text, word error rate (WER), and MSE of proposed LR.

## 6.2 LR FOR SPEECH RECOGNITION

To demonstrate LR's wide applicability, in addition to computer vision, we also demonstrate its performance against a speech recognition attack. As target model, we use Wav2vec (Schneider et al., 2019) model, which is trained on LibriSpeech (Panayotov et al., 2015) dataset. Model is attacked with FGSM (Goodfellow et al., 2014). LR achieves an average AUC of $0.99$. Figure 6 illustrates some attacked and clean samples from LibriSpeech, along with the MSE values of LR. As shown in the figure, LR can even detect stealthy attacks which do not raise the word error rate (WER) much while distorting the recognized speech. Remarkably, while the WER of the first adversarial example in the figure is lower than the WER of the second clean example ($0.4$ vs. $0.5$), the LR loss for this stealthy adversarial sample is more than ten folds greater than that of the second clean example ($1.6$ vs $0.13$).

## 7 CONCLUSION

Although there are effective defense methods for specific model-attack combinations, their success do not generalize to all popular models and attacks. In this work, we filled this gap by introducing Layer Regression (LR), the first universal method for detecting adversarial examples. In addition to its universality, LR is much more lightweight and faster than the existing defense methods. By analyzing the common objectives of attacks and the sequential layer-based nature of DNNs, we proved that the impact of adversarial samples is more on the final layer than the first layer. Motivated by this fact, LR trains a multi-layer perceptron (MLP) on clean samples to estimate the feature vector using a combination of outputs from early layers. The estimation error of LR for adversarial samples is typically much higher than the error for clean samples, which enables an average AUC of $0.982$ and $0.970$ on the ImageNet and CIFAR-100 datasets across 6 models and 7 attacks. With extensive experiments, we showed that LR outperforms the existing defenses in image recognition by a wide margin and also provides highly effective detection performance in distinct domains, namely video action recognition and speech recognition. One caveat that needs to be studied in future works is the possibility of training an attack model that can learn to fool the target model and LR together.

## 8 REPRODUCIBILITY STATEMENT

We provide detailed information about our experimental settings and training methods in Section 4 and Appendix C. In section 3.1 we offer a step-by-step explanation of our detection algorithm, In Appendix D, a PyTorch-like pseudo code is provided. Furthermore, at the time of paper submission, we also anonymously provide one trained detector for a specific target model, along with the code and instructions for reproducing the results for that model on ImageNet. All trained models, code, and the project page will be shared publicly after the double-blind review period.

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

## A    MORE RELATED WORKS FROM OTHER DOMAINS

Adversarial examples are studied on several other domains including video action recognition Pony et al. (2021), traffic sign recognition Hsiao et al. (2024), object detection Zhao et al. (2019), video anomaly detection Mumcu et al. (2022), and speech recognition. (Żelasko et al., 2021). For instance, flickering attack (Flick) Pony et al. (2021) tries to attack action recognition models by changing the RGB stream of the input videos. Hsiao et al. (2024) investigates the effects of natural light on traffic signs and use it to generate adversarial examples.

Advit Xiao et al. (2019) is a detection method introduced for videos. It generates pseudo frames using optical flow and evaluates the consistency between the outputs for original inputs and pseudo frames to detect attacks. Another defense method, Shuffle Hwang et al. (2023), tries to increase the robustness of action recognition models by randomly shuffling the input frames. Mumcu & Yilmaz (2024b) proposed to train a lightweight VLM and use it for adversarial traffic sign detection. In speech recognition, defenses like denoising or smoothing are studied against adversarial examples (Żelasko et al., 2021).

## B    MORE DETAILS ON EXPERIMENTAL SETTINGS

From the validation set of ImageNet (Russakovsky et al., 2015), we used 40,000 images to train the detectors and the remaining 10,000 for testing. For CIFAR-100 (Krizhevsky et al., 2009) has 100 classes and there are 500 training images and 100 testing images per class, resulting in a total of 50,000 training and 10,000 test images. During the tests, only the correctly classified images by the target models were used. Total number for test images for each specific model and dataset is given in Table 5.

|  | Vgg19 | ResNet50 | Inceptionv3 | ViT | DeiT | LeViT |
|---|---|---|---|---|---|---|
| ImageNet | 4257 | 7701 | 7410 | 8457 | 7907 | 7639 |
| CIFAR-100 | 7045 | 8315 | 8089 | 8544 | 8667 | 8359 |

Table 5: Number of test images used for each model.

VLAD (Mumcu & Yilmaz, 2024c) was originally proposed for video recognition attacks, with the initial implementation designed for videos consisting of 30 frames. In the experiments conducted in Section 4, we used the official VLAD implementation. However, instead of averaging the scores across 30 frames, we applied it to a single image.

## C    SELECTED SUBSET VECTORS DURING LR TRAINING

A specific subset of layer vectors $a_r$, as described in equation 7, is chosen for each target model that is used during the experiments in Section 4. For the models, Pytorch Image Models (timm) (Wightman, 2019) is used.

Then, For

1. Resnet50: Layers have the name *conv2* were filtered, then among 15 *conv2* layers, 5th, 8th and 13th layers, for all ImageNet and CIFAR-100 tests

2. InceptionV3: Layers have the name *conv* were filtered, then among 94 *conv* layers, 15th, 25th and 35th layers, for all ImageNet and CIFAR-100 tests

3. Vgg19: Layers which have the name *features* were filtered, then among 37 *features* layers, 8th, 13th and 17th layers, for all ImageNet and CIFAR-100 tests

4. ViT: Layers which have the name *attn.proj* were filtered, then among 23 *attn.proj* layers, 8th, 13th and 17th layers, for all ImageNet and CIFAR-100 tests

5. DeiT: Layers which have the name *attn.proj* were filtered, then among 24 *attn.proj* layers, 8th, 13th and 17th layers for ImageNet tests, 5th, 6th, 7th layers for CIFAR-100 tests

6. LeViT: Layers which have the name *attn.proj* were filtered, then among 12 *attn.proj* layers, 3th, 5th and 7th layers for ImageNet tests, 5th, 6th, 7th layers for CIFAR-100 tests

were used as $a_r$ subsets. Before concatenating the $a_r$ vectors, a specific slicing function for each vector is applied as described in equation 8. The specific slicing functions for each corresponding $a_r$ are detailed below:

1. Resnet50: $[:5,:28,:28]$, $[:50,:7,:7]$ and $[:10,:14,:14]$.
2. InceptionV3: $[:3,:35,:35]$, $[3:,35:,35]$ and $[:3,:17,:17]$.
3. Vgg19: $[:5,:25,:25]$, $[:5,:25,:25]$ and $[:5,:25,:25]$.
4. ViT: $[:,:4:200]$, $[:,:4:200]$ and $[:,:4:,200]$.
5. DeiT: $[:,:4,:200]$, $[:,:4,:200]$ and $[:,:4,:200]$.
6. LeViT $[:4,:14:,14]$, $[:14,:7,:7]$ and $[:14,:7,:7]$.

The vector v is generated by concatenating the sliced $a_r$ vectors as described in equation 9. After acquiring $v$ for a model, an MLP with two hidden layers trained to minimize the MSE loss between $v$ and the feature vector $a_{n-1}$. Adam optimizer with $3 \cdot 10^{-4}$ learning rate is used for the training.

# D    PSEUDO-CODE FOR LR

In section 3.1, we explain our detection algorithm in details. Here, in Algorithm 1, we also provide a PyTorch Paszke et al. (2017) like pseudo code for LR.

---

Algorithm 1: Linear Regression (LR)

---

**Input:** input $x$, selected subset of vector layers $a_r$, slicing functions $s$, DNN model $g$, feature vector $a_{n-1}$, LR detector $m$, threshold $h$.
**Output:** detection result $d$.
1: DNN $g$ takes the input $x$ as $g(x)$
2: Each layer vector $a_r$, process with corresponding slicing function in $s$
3: **for** $a_i$ in $a_r$ **do**
4:      $s_r \leftarrow s_i(a_i(x))$
5: **end for**
6: $v \leftarrow torch.cat(s_r)$
7: feed $v$ into a detection model $m$, such that: $m(v)$
8: calculate the MSE loss between $m(v)$ and $a_{n-1}$:
9: $l \leftarrow MSE(m(v), a_{n-1})$
10: **if** training **then**:
11:      $l.backward()$
12:      $optimizer.step()$
13: **else**
14:      **if** $l > h$ **then**
15:          $d \leftarrow$ adversarial
16:      **else**
17:          $d \leftarrow$ not adversarial
18:      **end if**
19:      **return** $d$
20: **end if**

---

# E VISUALIZED ADVERSARIAL EXAMPLES

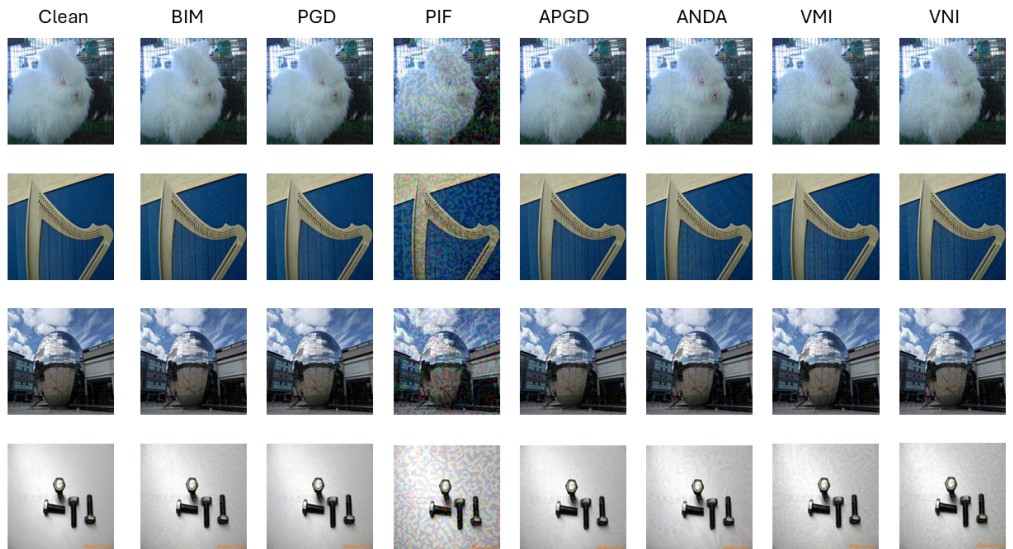

Figure 7: Adversarial examples generated with BIM Kurakin et al. (2018), PGD Madry et al. (2017), PIF Gao et al. (2020), APGD Croce & Hein (2020), ANDA Fang et al. (2024), VMI and VNI Wang & He (2021) by attacking target model ResNet50 He et al. (2015). First column represents the clean samples.

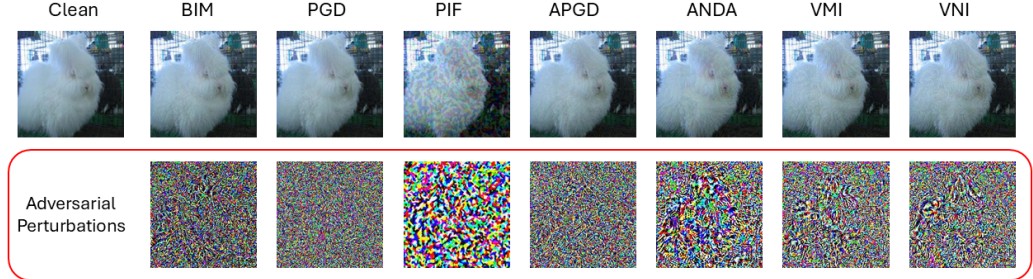

Figure 8: Visualization perturbations generated with BIM Kurakin et al. (2018), PGD Madry et al. (2017), PIF Gao et al. (2020), APGD Croce & Hein (2020), ANDA Fang et al. (2024), VMI and VNI Wang & He (2021) by attacking target model ResNet50 He et al. (2015). First row shows the clean sample and corresponding adversairal example for each attack. Second row demonstrates the noises added to the clean sample by each attack.

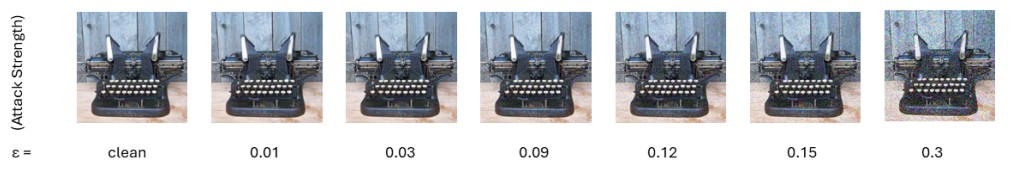

.

Figure 9: Effects of different attack strength values on a sample. The attack strength parameter $\epsilon$ is increased in the direction of the arrow. In the first sample, there is no attack. Samples are generated with adversarial attack PGD Madry et al. (2017) and target model ResNet50 He et al. (2015)

|    | FGSM | PGD | Patch | Light ‖ Average |
|----|------|-----|-------|-----------------|
| LR | 0.97 | 0.99 | 0.95 | 0.94 ‖ 0.96 |

Table 6: Detection AUC of LR against four attacks targeting ResNet50 in the traffic sign recognition task.

## F  ADDITIONAL ABLATION STUDY: LR IN TRAFFIC SIGN DETECTION

As an additional experiment, we implement LR against attacks that target traffic sign recognition. In this study, ResNet50 is used as target model and attacked with FGSM(Goodfellow et al., 2014), PGD(Madry et al., 2017), Light(Hsiao et al., 2024) and Patch (Ye et al., 2021) attacks. In Table 6, we show that LR achieves an average AUC score of 96%, which further proves the applicability and success of our detection method across different domains.

## G  DETAILS OF ABLATION STUDY & OTHER DOMAIN EXPERIMENTS

For Section 6.1, the experimental settings detailed in Mumcu & Yilmaz (2024c) is followed: "A subset of Kinetics-400 Kay et al. (2017) is randomly selected for each target model from the videos that are correctly classified by the respective model. For each subset, the total number of the videos are between 7700 and 8000 and each class has at least 3, at most 20 instances. An adversarial version of the remaining 20% portion is generated with each adversarial attack, in a way that they cannot be correctly classified by the models. Then the adversarial set is used for evaluation along with the clean versions." Similarly to main experiments, we trained an MLP with 2 hidden layers with the same training hyper-parameters described in Appendix C. For MVIT (Fan et al., 2021) and CSN (Tran et al., 2019) models, layers which have the name *conv_a* and *attn.proj* were filtered respectively. While for both of the models 3th 5th and 7th layers used to form subset $a_r$, for CSN (Tran et al., 2019) additional layers 4th, 6th and 8th were also used. $[:,:4,:200]$ and $[:,:4,:3,:7,:7]$ were used as slicing functions for MVIT (Fan et al., 2021) and CSN (Tran et al., 2019) respectively.

For Section 6.2, an MLP with 2 hidden layers with the same training hyper-parameters described in Appendix C is trained. For Wav2vec Schneider et al. (2019) model, layers which have the name *attention.dropout* were filtered, and 3th layer used to form $a_r$. As slicing function $[:,:4,:20,:10]$ is selected. While 500 sound clips from LibriSpeech dataset used for experiments, 2000 sound clips saved for training.

For Appendix F, we use the experimental settings as demonstrated in Mumcu & Yilmaz (2024b) and use the GTSRB Stallkamp et al. (2012) dataset which includes 43 classes of traffic signs, split into 39,209 176 training images and 12,630 test images. LR detector is trained with the same settings that are specified for Resnet50 in Appendix C.

