# OpenReview forum: "Detecting Adversarial Examples"
_ICLR.cc/2025/Conference — ICLR 2025 Conference Withdrawn Submission_

### Official Review · Reviewer_naEt · 2024-10-20

**Soundness:** 1
**Presentation:** 2
**Contribution:** 1
**Rating:** 1
**Confidence:** 5

**Summary:**

This paper proposes a universal method for detecting adversarial examples by analyzing the layer outputs of deep neural networks. The authors claim to have theoretical justifications and experiments to demonstrate the effectiveness of their method on any DNN architecture and its applicability across multiple domains, including image, video, and audio. Empirical experiments are done on CIFAR-100 and ImageNet.

**Strengths:**

I tried, but it's difficult to write down any points that deserve to be called Strengths for an ICLR-submitted paper.

**Weaknesses:**

The weaknesses of this paper include:

- **The proof of Theorem 1 is wrong.** The proof bases on that "Finally, the perturbation aligned with DNN weights is amplified as it sequentially moves through the DNN layers (Goodfellow et al. 2014)", which is an *empirical observation*, not a theoretical conclusion. I was shocked that the authors treat an empirical observation as a formal Theorem, and it's easy to construct a counter-example DNN that violate Theorem 1.

- **Non-adaptive evaluations.** In Line 290-292, the authors claim that "We consider the most challenging case for the detector, the white-box attack scenario, in which the attacker has full access to the classification model". This is NOT a white-box scenario for detection, because *the authors did not assume that the attacker has access to the detection model*. The authors should design adaptive attacks [1,2], where the attacker has full access to both the classification and detection models.

- **False AUC values.** As indicated in [3], a detection AUC value can be converted into a classification defense accuracy. Up to now, the state-of-the-art defenses as listed on RobustBench [4] is less than 45% acc (CIFAR-100), which cannot match the mostly >0.99 AUC values reported in this paper.

- **The baselines are weak.** The majority of defense baselines in this paper are simple input processing methods like JPEG, which is far from a fair comparison for a paper submitted in 2024.


References:\
[1] Adversarial examples are not easily detected: Bypassing ten detection methods. AISec 2017\
[2] Obfuscated gradients give a false sense of security: Circumventing defenses to adversarial examples. ICML 2018\
[3] Detecting Adversarial Examples Is (Nearly) As Hard As Classifying Them. ICML 2022\
[4] https://robustbench.github.io/

**Questions:**

Some questions:

- In the abstract, the authors claim that their method is "applicable across different domains, such as image, video, and audio." However, I can only find experiment results on CIFAR-100 and ImageNet, so where are the results on video and audio?

- The authors claim that their LR method can detect unseen attacks. Where are related experiments? For example, can LR method trained on $\ell_{\infty}$ attacks detect $\ell_{2}$ or $\ell_{0}$ attacks?

- Why there is no experiments on CIFAR-10? CIFAR-10 is the most commonly used dataset in the adversarial literature.

---

> ### Author Response · Authors · 2024-11-18
>
> We tried, but it's difficult to respond to a reviewer who clearly did not read the paper and did not even understand the main topic of the paper.
>
> -The results for video and speech recognition are presented in chapter 6. There is a section dedicated to this which is called “APPLICABILITY IN OTHER DOMAINS”.
>
> -We clearly defined our evaluation metric; it cannot be compared with other results from robustbench website which is on robust classification and not detection.

---

> ### Comment · Reviewer_naEt · 2024-11-21
>
> The authors seem unable to respond to my concerns in the Weaknesses section:
>
> - **The proof of Theorem 1 is wrong**
>
> - **False AUC values**
>
> - **The baselines are weak**
>
> During the rebuttal, I saw the authors' efforts to design adaptive attacks, and I appreciate them bringing up Chapter 6. I have to admit that I lost interest in this paper when I discovered that **the authors treat an empirical observation as a formal theorem**.
>
> I strongly suggest the authors to thoroughly read ``Detecting Adversarial Examples Is (Nearly) As Hard As Classifying Them. ICML 2022 Oral`` and then they will understand why their AUC values are false.

---

> > ### Author Response · Authors · 2024-11-26
> >
> > Thank you for confirming that you did not fully read the paper. Our AUC results are not false. They are obtained on a dataset in which attacks successfully deceive the target models. The dataset consists of clean and perturbed versions of the images. Dataset details are given in the paper. The reported AUC results are based on the false positive rate (declaring a clean image as perturbed) and true positive rate (successfully detecting a perturbed image) on this dataset considering the attacks and target models stated in the paper.
> >
> > The cited paper does not show that our results are wrong. It only shows that there is an impractical robust classifier which can find the correct class of the image at the same accuracy with half attack strength. Attack detection and robust classification are two different problems. Although the cited paper tries to establish a link between the two problems, their results do not invalidate the importance of the detection problem. A successful detector practically and efficiently detects malicious images. The fact that there might be an impractical theoretical robust classifier implied by the detector does not make the detector useless or its results false. An obvious example showing the fundamental difference between the attack detection and robust classification problems is the increasing attack strength case. When the attack strength increases, the attack detection problems becomes easier for a successful detector, as illustrated in the paper in Figure 5; whereas, the robust classification problem becomes harder for any classifier, as shown in Table 1 in the cited paper.
> >
> > Robust classification research continues its mission to develop efficient practical robust classifiers that can approach the high accuracies implied by a successful practical detector. If the current robust classification literature does not have a practical and highly accurate robust classifier, this definitely does not show the practical detectors are doing something wrong. Again, with the considered attacks and target models on the explained datasets, our practical detector achieves the reported AUCs. The reported results are not wrong.
> >
> > We strongly suggest the reviewer to thoroughly read the attack detection literature to understand the fundamental difference between the two problems: attack detection and robust classification. Then, they might understand why their mindset is wrong.

---

### Official Review · Reviewer_MCAB · 2024-10-26

**Soundness:** 1
**Presentation:** 3
**Contribution:** 1
**Rating:** 3
**Confidence:** 5

**Summary:**

This paper introduces a technique to detect adversarial examples. The defense works by studying the activations of the model to notice suspicious patters that are not present in the clean data. It presents evidence of its efficacy by evaluating against a suite of attacks, and compares to several prior defenses from the literature.

**Strengths:**

Defending against adversarial examples is an important and interesting challenge.

**Weaknesses:**

Unfortunately, it appears unfamiliar with the (vast) literature on this topic and the paper does not present convincing evidence that it will be robust to adaptive attacks.

I would particularly recommend the authors begin by reviewing Carlini & Wagner "Adversarial Examples Are Not Easily Detected: Bypassing Ten Detection Methods", and Tramer "Detecting Adversarial Examples Is (Nearly) As Hard As Classifying Them". The latter paper, in particular, shows that the results claimed here would imply a nearly perfectly robust classifier. Then, I would recommend the authors read related papers on detecting adversarial examples ["The Odds are Odd", "Asymmetrical Adversarial Training"]. After this, it could be then useful to study how these defenses were broken in "On adaptive attacks to adversarial example defenses".

What this paper presents---an evaluation against a large set of fixed attacks---is not sufficient for arguing adversarial robustness, and I can not recommend acceptance at this point.

(I would also recommend changing the title of this paper. There are 50+ papers on detecting adversarial examples.)

**Questions:**

Do you believe this paper will be robust to adaptive attacks, as discussed above?

---

> ### Author Response · Authors · 2024-11-18
>
> **What this paper presents---an evaluation against a large set of fixed attacks---is not sufficient for arguing adversarial robustness, and I can not recommend acceptance at this point. (I would also recommend changing the title of this paper. There are 50+ papers on detecting adversarial examples.) Do you believe this paper will be robust to adaptive attacks, as discussed above?**
>
> Please see our response in the common response section regarding the adaptive attacks.

---

### Official Review · Reviewer_xCZW · 2024-10-31

**Soundness:** 2
**Presentation:** 2
**Contribution:** 2
**Rating:** 5
**Confidence:** 5

**Summary:**

This paper proposes a detection method named Layer Regression (LR), which assumes that the impact of adversarial examples on the final layer is much larger than the initial layer. The proposed LR can achieves high detection performance, and experimental results show that it can be applied in different tasks.

**Strengths:**

1. The motivation of the proposed LR is clearly stated.
2. According to the experimental results, LR detects adversarial examples with high efficiency.
3. Experiments in other domains are implemented to prove the universality of LR.

**Weaknesses:**

1. The detection baselines are not strong enough. Some strong baselines, like [1][2][3], are not included, which makes the experimental results less convincing.
2. There is a lack of adaptive attack against LR, and the adaptive attack is important to evaluate the detection performance.
3. According to Section C in the Appendix, the subset of layer vectors needs to be selected for each model, and the dataset may sometimes influence the choice of layers, reducing the practicality of LR.

[1] Tian, Jinyu, Jiantao Zhou, Yuanman Li, and Jia Duan. "Detecting adversarial examples from sensitivity inconsistency of spatial-transform domain." In Proceedings of the AAAI Conference on Artificial Intelligence, vol. 35, no. 11, pp. 9877-9885. 2021.
[2] Yang, Yijun, Ruiyuan Gao, Yu Li, Qiuxia Lai, and Qiang Xu. "What you see is not what the network infers: Detecting adversarial examples based on semantic contradiction." NDSS 2022.
[3] Zhang, Shuhai, Feng Liu, Jiahao Yang, Yifan Yang, Changsheng Li, Bo Han, and Mingkui Tan. "Detecting adversarial data by probing multiple perturbations using expected perturbation score." In International conference on machine learning, pp. 41429-41451. PMLR, 2023.

**Questions:**

1. What will the detection performance of LR be when facing targeted attacks?

---

> ### Author Response · Authors · 2024-11-18
>
> **The detection baselines are not strong enough. Some strong baselines, like [1][2][3], are not included, which makes the experimental results less convincing.**
>
> Thanks for the comment. We will expand our comparisons with the other detection methods, including the proposed ones.
>
>
> **There is a lack of adaptive attack against LR, and the adaptive attack is important to evaluate the detection performance.**
>
> Please see our response in the common response section regarding the adaptive attacks.
>
>
> **According to Section C in the Appendix, the subset of layer vectors needs to be selected for each model, and the dataset may sometimes influence the choice of layers, reducing the practicality of LR.**
>
> Thank you for your insight on the layer selection and reading our paper through appendix. In this work we tried to understand the characteristics of LR including the effects of layer selection. We showed that using only early layers as input affect the performance negatively. We shared the subset of layer vectors to show what combinations are used. We plan to implement an automated layer selection mechanism in the future. We will add this discussion as a future work to our paper.
>
>
> **What will the detection performance of LR be when facing targeted attacks?**
>
> Thank you for your suggestion on targeted attacks. During our preliminary experiments for adaptive attacks, we also conducted a targeted pgd attack against ViT where we found 0.97 as detection AUC. The detection AUC is 0.98 for untargeted pgd attack against ViT, as reporte in the paper. We expect LR to be successful against targeted attacks too.

---

> > ### Comment · Reviewer_xCZW · 2024-11-25
> >
> > Thank you for your response to all the comments. According to the experimental results of the adaptive attack, when λ=1, ASR is 0.954, the AUC score drops to 0.646, which needs to be enhanced. Besides, the attack strength of the adaptive attack is not mentioned. Meanwhile, the problems, like the lack of strong baseline comparisons, and the unaddressed layer selection issue, remain to be solved. The authors have proposed solutions for some of these issues, but these are mostly in the form of future work. As a result, my score remains unchanged.

---

### Official Review · Reviewer_EnGb · 2024-11-01

**Soundness:** 1
**Presentation:** 3
**Contribution:** 3
**Rating:** 3
**Confidence:** 4

**Summary:**

This study proposes a universal and lightweight detection method for adversarial examples to defend deep neural networks from the threat of adversarial examples. The proposed detector is trained to predict the difference in feature vectors of benign inputs among layers. When the loss of the input is higher than a pre-defined threshold, the detector treats the input as an adversarial example. The effectiveness of the proposed detector is justified through theoretical analysis and extensive experiments.

**Strengths:**

- The proposed method empirically shows meaningful improvement in detection AUC compared to other baseline defenses among various architectures.
- The motivation is well explained and partially justified through theoretical analysis.

**Weaknesses:**

- The proof of theorem 1 seems to miss the important assumption in the referred paper (Goodfellow et al., 2014). Goodfellow et al. claim that the perturbation is linearly amplified as it moves through linear models, but there are no theoretical results for nonlinear models.
- The effectiveness of the proposed detector could be further emphasized by investigating their detection performance against attacks that produce adversarial examples with minimal perturbation norms because the tested attacks are limited to maximize attack success rates under a given attack budget.

**Questions:**

- Does the theorem 1 hold for arbitrary deep neural networks? I would like to show you a strict mathematical proof of theorem 1.
- Is the proposed method able to detect adversarial examples with a minimal norm of adversarial perturbation?

---

> ### Author Response · Authors · 2024-11-18
>
> **The proof of theorem 1 seems to miss the important assumption in the referred paper (Goodfellow et al., 2014). Goodfellow et al. claim that the perturbation is linearly amplified as it moves through linear models, but there are no theoretical results for nonlinear models. Does the theorem 1 hold for arbitrary deep neural networks? I would like to show you a strict mathematical proof of theorem 1.**
>
> Thank you for the feedback. We will more carefully state our assumptions in the theoretical result.
>
>
> **The effectiveness of the proposed detector could be further emphasized by investigating their detection performance against attacks that produce adversarial examples with minimal perturbation norms because the tested attacks are limited to maximize attack success rates under a given attack budget. Is the proposed method able to detect adversarial examples with a minimal norm of adversarial perturbation?**
>
>
> Yes, our method is able to detect adversarial examples with minimal norm of perturbation that can still deceive the target classifiers. In Figure 5, we analyzed minimizing the perturbation norm up to the point where it can deceive the target models on a sizable set of images.

---

### Official Review · Reviewer_Cq5w · 2024-11-04

**Soundness:** 2
**Presentation:** 2
**Contribution:** 1
**Rating:** 3
**Confidence:** 4

**Summary:**

This paper proposes Layer Regression (LR), a universal and lightweight method for detecting adversarial examples in DNNs. The key innovation lies in analyzing the changes in DNN's internal layer outputs rather than focusing on input modifications or output comparisons like previous approaches. The authors provide theoretical justification through a theorem showing that adversarial impacts are stronger in the final layers compared to the initial layers.

**Strengths:**

1. **Theoretical Foundation.** Mathematical proof supporting the core concept and has a clear theoretical justification for why the method works.

2. **Lightweight:** Using a relatively small MLP for regression makes LR computationally efficient, and suitable for real-time detection.

**Weaknesses:**

See questions.

**Questions:**

1. **Input Selection Heuristic:** The selection of early layer activations for the regression model's input seems somewhat arbitrary. While the paper mentions a trade-off between proximity of clean and adversarial inputs and the accuracy of the estimator, a more principled approach for input selection would strengthen the method.
2. **Lack of Robustness Analysis:** The paper primarily focuses on white-box attacks. The performance against more realistic ***black-box attacks*** is not evaluated. Furthermore, the robustness of LR itself against ***adaptive attacks***, where the adversary is aware of the defense mechanism, is not discussed. An attacker could potentially craft perturbations that minimize the change in the selected early layer activations while still maximizing the classification loss.
3. **Overfitting Potential:** Training the MLP on clean data only might lead to overfitting and poor generalization to unseen adversarial examples, especially considering the high dimensionality of the feature vectors.
4. **Hyperparameter sensitivity:** The method requires setting a threshold value, which could be sensitive to different scenarios.
5. **Inadequate Evaluation:** The baseline methodology used is outdated and not compared to the most recent SOTA **[R1], [R2]**.

**[R1]** What You See in Not What the Network Infers: Detecting Adversarial Examples Based on Semantic Contradiction.
**[R2]** Detecting adversarial data by probing multiple perturbations using expected perturbation score.

---

> ### Author Response · Authors · 2024-11-18
>
> **Input Selection Heuristic: The selection of early layer activations for the regression model's input seems somewhat arbitrary. While the paper mentions a trade-off between proximity of clean and adversarial inputs and the accuracy of the estimator, a more principled approach for input selection would strengthen the method.**
>
> Thank you for pointing out activation layer selection. In this work, we showed that using a mixture of layers as input improves the performance compared to using only early layers . A more detailed and principled selection is left as a future work.
>
> In this work, we tried to understand the characteristics of LR including the effects of layer selection. In appendix, we shared the subset of layer vectors to show what combinations are used and became successful. We plan to implement an automated layer selection mechanism in the future. We will add this discussion as a future work to our paper.
>
>
>
> **Lack of Robustness Analysis: The paper primarily focuses on white-box attacks. The performance against more realistic black-box attacks is not evaluated. Furthermore, the robustness of LR itself against adaptive attacks, where the adversary is aware of the defense mechanism, is not discussed. An attacker could potentially craft perturbations that minimize the change in the selected early layer activations while still maximizing the classification loss.**
>
> Thank you for pointing out black-box attacks and adaptive attacks. Please see our response in the common response section regarding the adaptive attacks. We didn’t conduct experiments with black-box attacks since they are considered to be easier to defend.
>
>
> **Overfitting Potential: Training the MLP on clean data only might lead to overfitting and poor generalization to unseen adversarial examples, especially considering the high dimensionality of the feature vectors.**
>
> We didn’t observe overfitting during our experiments. That is because of the well-defined datasets that we used and the size of the datasets. However, the reviewer might be right in case of the usage of other datasets which are more prone to overfitting. Those cases should be investigated individually and common ML methods for avoiding overfitting should be used.
>
>
> **Hyperparameter sensitivity: The method requires setting a threshold value, which could be sensitive to different scenarios.**
>
> All practical detectors use a threshold for the final decision. In practice, the detection threshold is typically set on a validation set to satisfy a given false alarm rate for different scenarios.
>
> **Inadequate Evaluation: The baseline methodology used is outdated and not compared to the most recent SOTA [R1], [R2].**
>
> Thanks for the comment. We will expand our comparisons with the other detection methods, including [R1] and [R2].

---

### Author Response · Authors · 2024-11-18

We thank the reviewers for their valuable comments and suggestions. We see that LR’s unknown performance under the adaptive attack setting is a common concern. Therefore, we will respond to it in this section. We design LR as a universal detection mechanism which is compatible with any DNN. For this reason, while designing our experiments, we focused on using as many different target models as possible and employed various white-box attacks to demonstrate the LR's effectiveness across different model-attack settings and different domains.

However, we agree that adaptive attacks have been investigated more in recent years, and it is important to explore the limits of LR under adaptive attack settings. We conducted a new experiment to show LR’s performance against an adaptive attack.

Following the approach demonstrated in [1] we integrate the LR loss function L_LR in the loss function of pgd attack:

$L_{PGD} = L_{classifier} - \lambda \cdot L_{LR}$

Note that, the sign of LR’s loss function is (-), differently from [1]. Since LR identifies an input as clean when the loss is low, in the adaptive attack scenario, the attacker wants to minimize the LR loss to avoid our detection while maximizing the classifier loss.

Using ViT as the target model and setting $\lambda$ as 1 we obtain the LR detection AUC as 0.65. We observed a similar drop in the performance with [1].

One possible way of improving the adaptive attacks’ success is to increase $\lambda$. However, this also leads to a decrease in the attack success rate (ASR). The table below shows the change in the attack success rate and LR’s detection AUC. λ=0 means the attack is not adaptive. Note that, in this experiment, attack success rate is the percentage of adversarial examples generated by adaptive targeted PGD that successfully cause ViT to misclassify the input.




| $\lambda$ | | 0 | 1 |5 | 10 |50 |
|--------------|-|--------------|--------------|--------------|--------------|--------------|
| ASR | | 0.97 | 0.954 | 0.612 | 0.221 | 0.002 |
| Detection AUC | | 0.977 | 0.646 | 0.356 | 0.149 | 0.024 |


We focused on developing a universal detection method against static attacks in different domains (image, video, audio). In the revised version, we will investigate the impact of adaptive attacks on our detector in detail.

[1]  Yang, Yijun, Ruiyuan Gao, Yu Li, Qiuxia Lai, and Qiang Xu. "What you see is not what the network infers: Detecting adversarial examples based on semantic contradiction." NDSS 2022

---

### Note · Authors · 2024-11-26

I have read and agree with the venue's withdrawal policy on behalf of myself and my co-authors.